# Spatial Distribution Characteristics of Microbial Mineralization in Saturated Sand Centrifuge Shaking Table Test

**DOI:** 10.3390/ma15176102

**Published:** 2022-09-02

**Authors:** Zhiguang Han, Jianzhang Xiao, Yingqi Wei

**Affiliations:** 1Department of Civil Engineering, Henan University, Kaifeng 475004, China; 2State Key Laboratory of Simulation and Regulation of Water Cycle in River Basin, China Institute of Water Resources and Hydropower Research, Beijing 100048, China

**Keywords:** saturated sand, centrifuge, vibration table, microbial grouting, mineralization, spatial distribution

## Abstract

Calcium carbonate induced by microorganisms can quickly fill and cement sand particles, thereby effectively reducing the potential for the liquefaction of sand. This process could represent a new green approach to the liquefaction treatment of saturated sand and has good prospects for application. However, owing to the diversity of microbial activities and the heterogenous spatiotemporal distribution of bacterial nutrient seepage in sandy soil foundations, the resultant complex distribution of calcium carbonate deposition in a sandy soil foundation can lead to differences in solidification strength and improvement effect. To understand the influence of earthquake action on the liquefaction resistance of saturated sand treated by microorganisms, and to evaluate the effect of microbial technology on sand liquefaction prevention under dynamic load, this study simulated the dynamic stress conditions of saturated sand under shear waves, using the world′s first centrifuge shaking table (R500B), which realizes horizontal and vertical two-way vibration. On the basis of spatial heterogeneity of microbial mineralization after centrifuge shaking table tests, the effect of microbial strengthening on liquefied sand was analyzed, and the spatial distribution of calcium carbonate mineralization was examined. The results showed that the distribution of microorganisms in the solidified soil exhibited obvious spatial heterogeneity with a significant edge effect. Although microbial mineralization effectively improved the liquefaction resistance of saturated sand, a sudden change in the process of calcium carbonate deposition altered the cementation of the sand with depth. Moreover, the curing strength had obvious complexity and uncertainty that directly affected the shear stiffness of the soil under dynamic load, and this constitutes one of the reasons for the degradation of shear stiffness of sand during liquefaction. The derived conclusions could be used as a reference for engineering applications of microbial treatment of a liquefiable sandy soil foundation.

## 1. Introduction

Sand liquefaction induced by an earthquake is a common sudden geological event. It is one of the causes of potentially fatal earthquake-related damage, such as the uneven settlement of building foundations and structural damage. Research and engineering circles in China and internationally have long attached great importance to this phenomenon. With the 2008 Wenchuan earthquake in China, large-scale severe liquefaction occurred [1], whereby the deep soil (below 20 m) of the overburden was liquefied in some areas [2,3]. This resulted in the destruction of a large number of buildings, bridges, underground pipelines, and other infrastructure and led to serious economic losses. Some shield tunnels in the metro systems of certain cities, e.g., Nanjing and Shenzhen, as well as in the Taipei Rapid Transit System, all encounter liquefiable strata [4,5] and are, therefore, at risk of structural damage caused by soil liquefaction. In comparison with the problems of low efficiency and environmental pollution associated with the liquefaction treatment of sand by using traditional physical or chemical methods, microbial grouting reinforcement technology is a new foundation treatment technology with a short construction period, low energy consumption, and obvious reinforcement effect [6,7]. Through the growth, migration, and reproduction of microorganisms in soil pores, the technology induces the formation of calcium carbonate crystals to quickly fill and cement surrounding sand particles [8,9,10,11,12,13,14]. This process increases the strength and stiffness of the foundation, reduces the likelihood of foundation damage caused by liquefaction sensitivity and dynamic load, and has the potential to replace traditional foundation treatment methods such as mechanical methods and chemical grouting [15,16]. Thus, it has wide-ranging engineering prospects in terms of application to liquefiable sand layers, slope stability, and subgrade reinforcement [17].

A centrifuge shaking table can realistically simulate the stress conditions of a prototype stress field, accurately reproduce the dynamic response of the prototype under the actual stress conditions, overcome the influence of simplified assumptions on the soil properties in theoretical calculation, and reduce the problems of high cost and long testing times in prototype observation [18]. Consequently, it is the most effective method and test technology available for the study of sand liquefaction [19,20,21,22,23].

The microbial grouting method is an innovative approach that is mainly used to improve the shear strength and stiffness of sandy soil, using the growth, migration, and propagation of microorganisms in porous media. Different from calcium carbonate synthesized by using general chemical methods, calcium carbonate crystal particles formed by microorganisms are small, have different shapes, and have strong cementation and filling effects. They can cement loose sand particles into a unit with certain mechanical properties that can substantially increase the strength and bearing capacity of a soft sand foundation and reduce its permeability. The strength and reaction speed of microbial solidified sand are controllable, and application of the technique has the advantages of being rapid and efficient, using non-toxic products, and having a clear mechanism. Thus, it has become the leading technology used for treatment of liquefiable sand. Today, with increasing ecological awareness, microbial rock/soil modification technology demonstrates strong competitiveness [24].

As a new type of foundation treatment technology, microbial grouting reinforcement has promoted advances in the field of soil strength solidification and modification [25,26]. However, few relevant studies have examined the seismic dynamic characteristics and response of saturated sand, and centrifuge shaking table model tests of microbial treatment of a liquefied sand foundation have rarely been reported. Specifically, knowledge is lacking regarding the diversity of microbial activities and the uneven spatiotemporal distribution of bacterial nutrient seepage in a sand foundation. Such uneven distribution can result in complex distribution of calcium carbonate deposition in sandy soil foundations, resulting in spatially inconsistent solidification strength and improvement effect. To understand the influence of earthquake action on the liquefaction resistance of saturated sand treated by microorganisms, and to evaluate the effect of microbial technology on the prevention of sand liquefaction under dynamic loads, this study systematically investigated the spatial distribution characteristics and distribution rules of microbial mineralization at different depths in a sand foundation under earthquake action. To simulate the earthquake action, this study employed the world’s first centrifuge shaking table (R500B), which was developed by the China Institute of Water Resources and Hydropower Research. The research experience herein could support the practical application of microbial treatment in liquefiable sandy soil foundation engineering.

## 2. Materials and Methods

### 2.1. Test Materials

#### 2.1.1. Standard Sand

The sand used in the model testing was Jinjiang XiFeng standard sand. Table 1 lists the basic physical parameters of the sand in accordance with the standards for soil test methods (GB/T 50123-2019). The specific gravity (*G*_S_) of the sand is 2.61. The grading curve is shown in Figure 1, and the nonuniformity coefficient (*C*_u_) is 1.655.

#### 2.1.2. Bacterial Solution and Nutrients

The bacterial strain used in the experiment was *Sporosarcina pasteurii*, which is a type of chemotrophic bacteria. The cells were rod-shaped, 2–3 µm in length, 0.5–1.5 µm in diameter, Gram-positive, and the spores were round.

To prepare the microbial grouting fluid, optimized #136 medium was prepared by adding 20.0 g of yeast powder, 10.0 g of ammonium sulfate, and 1.0 mL of nickel chloride for every 1.0 L of water. Sodium hydroxide solution was added to adjust the pH of the medium to 9.0, and the medium was sterilized at 120 °C for 40 min. After inoculation, the bacterial culture was placed in a shaking incubator at 30 °C and shaken at 170 rpm for 24 h [27,28,29,30]. After oscillation, while kept at an indoor temperature of 22 °C, the measured biomass OD_600_ of the bacterial solution was 2.293 on average, and the urease activity was 2.0 ms/min/cm on average. The nutrient used in the test was Ca(CH_3_COO)_2_.

#### 2.1.3. Saturated Solution

In the centrifugal simulation test, the time-similarity ratio under dynamic action was 1/n, and under the condition of the saturated sample, the time-similarity ratio in the seepage process was 1/n^2^. To ensure that the pore pressure did not dissipate too quickly during the vibration process, the time-similarity ratio under the action of seepage was close to the similarity ratio under the action of the dynamic force of 1/n, making the simulation results more accurate. The saturated solution used in the test was hydroxypropyl methyl cellulose, with a mass concentration of 0.48% and a viscosity of 30 times that of pure water.

### 2.2. Test Equipment

The Lxj-4-450 large-scale geotechnical centrifuge model test machine and the R500B shaking table (Figure 2a) of the China Academy of Water Resources and Hydropower Research were used in the test. The R500B is the first centrifuge shaking table in China that can realize horizontal and vertical two-way vibration. It can apply the seismic load of any waveform under the condition of maximum centrifugal acceleration of 60 g. The maximum load is 440 kg. The seismic peak values that can be input for the horizontal and vertical planes are 30 and 20 g, respectively, and the frequency range is 0–400 Hz.

The model box comprised a stacked ring model box constructed from aviation ultra-high-strength light aluminum alloy (Figure 2b), with a mass of 124 kg and internal dimensions of 800 × 353 × 415 mm, corresponding to a prototype depth of 9.6 m. The model box was stacked with 15 layers in rectangular frames; each layer was 2.66 cm high, and a rubber layer was embedded between every two adjacent layers. To ensure tightness and to prevent soil particles from entering the stack, a rubber film with a thickness of 2.5 mm was laid inside the model box.

### 2.3. Grouting Scheme

The bacterial solution with high biological activity after laboratory mutagenesis was mixed with Ca(CH_3_COO)_2_ solution at a low concentration of 0.5 mol/L and then poured quickly into the sand sample. The microorganisms were then cultured in situ for 24 h in the sand sample. Subsequently, the nutrient salt of the mixed solution of Ca(CH_3_COO)_2_ and urea with the same concentration of 0.5 mol/L was added at a low speed.

To ensure uniform effect of the calcium carbonate produced in the process of microbial grouting, a peristaltic pump was used for two-way grouting.

### 2.4. Model Preparation

The model sample was constructed by using the sand rain method. According to the calibration relationship curve between falling distance and relative compaction (Table 2 and Figure 3), the falling distance of dry sand during the test was determined to be 28 cm. After grouting, the model was saturated with purified water for 24 h.

The preparation sequence of the microbial grouting model was as follows. The grouting pipe was pasted on the inner wall of the model box by using 14 pieces of double-sided adhesive tape attached at equal intervals on each long side. The top grouting hole was placed 3 cm below the sand surface, and all grouting holes were confirmed to be parallel. The dry sand model was constructed by using the sand rain method, and two peristaltic pumps were used to inject and discharge liquid through the grouting pipes to ensure that the grouting liquid flowed evenly throughout the model box as far as was possible. The intermittent grouting method was adopted, and the flow rate of the grouting and drainage was controlled at 10–12 mL/min. When the model was completely static, the liquid on the surface of the model was removed, and the measured density of the microbial grouting sand sample was 1.960 g/cm^3^.

### 2.5. Centrifuge Test g Value and Vibration Wave Input

After the model was prepared, it was installed on the R500B shaking table, and accelerometers were connected to the measurement system. During the model test, the acceleration of the centrifuge was controlled at 30 g, and the shaking table test was started once the centrifuge was stable.

Sine-wave loading was adopted in the test. The amplitude in the time domain and the normalized frequency response in the frequency domain of the loading signal are shown in Figure 4a, b, respectively. The maximum acceleration of the input sine excitation signal was 3.92 m/s^2^, and the frequency was 2.33 Hz.

## 3. Results and Discussion

To understand the influence of earthquake action on the liquefaction resistance of saturated sand treated by microorganisms, and to evaluate the effect of microbial technology on the prevention of sand liquefaction under dynamic load, the spatial distribution characteristics of microbial mineralization in the model box sand foundation were systematically analyzed, following the centrifuge shaking table test.

### 3.1. Apparent Distribution of Calcium Carbonate Deposition

Following the centrifugal vibration test with duration of 24 h, the formwork was removed. During formwork removal, the depth of the sand surface was determined to be less than 2 cm. The hard calcium carbonate sediment shell on the sand’s surface that was strengthened by microorganisms was already cracked and fragmented before vibration was applied (Figure 5a). Scant liquid was present at the sensor position on the top layer, and there were irregular calcium carbonate deposits around the sensor (Figure 5b).

When the depth of the sand was greater than 2 cm, no liquid was observed. The sand itself has a certain viscosity, and microbial action forms a continuous distribution of sediment with a certain strength (Figure 6).

In the horizontal plane, the calcium carbonate deposition in and around the bored grouting pipe in the model box was substantially greater than that in other parts (Figure 7a). During cleaning of the bored grouting pipe, it was also found that the calcium carbonate deposition induced by the microorganisms had accumulated at the bottom of the bored pipe, resulting in the blockage of some of the grouting holes (Figure 7b).

### 3.2. Sampling Method and Determination of Content

Samples were taken at 5 cm intervals downward from the sand surface, and a total of six layers were sampled. A grid sampling method was adopted for the sampling of each soil layer, as shown in Figure 8. The collected sand samples were placed into plastic zip-lock bags that were then sealed and placed in cold storage.

To determine the calcium carbonate content, a sample of sand was weighed and placed in an oven at 60 °C for 24 h. Then a 2 g sample of dried sand was placed into a 150 mL conical flask, and the total weight (M1) in grams was determined. Next, 100 mL of deionized water was poured into the conical flask, which was shaken, and then the Ca^2+^ concentration in the supernatant was determined by atomic absorption spectrometry (C1) mg/L. The supernatant was poured out, and the flask was weighed again (M2) in grams. Finally, the remaining sandy soil was dissolved in 100 mL of dilute hydrochloric acid and shaken, and the Ca^2+^ concentration in the supernatant (C2) mg/L was determined after the reaction was completed. The calcium carbonate content of the sand was calculated according to the relation [c2 × 0.1 − C1 × (M2 − M1) × 0.001]/0.4.

### 3.3. Spatial Distribution Characteristics of Microorganisms and Calcium Carbonate

The inverse distance weighted spatial interpolation method [31,32,33] was adopted to produce maps of the spatial distribution of calcium carbonate and microbial quantity.

The basic principle is as follows. Suppose a series of discrete points are distributed on a plane and the coordinate values are known to be Xi (i = 1, 2,…, n). Then U(x) can be interpolated by using the distance weighted value to satisfy the following relationship:(1)$$u\left(X\right)=\{\begin{array}{c} \frac{\sum_{i=1}^{N}w_{i}\left(X\right)u_{i}}{\sum_{i=1}^{N}w_{i}\left(X\right)},\;\;\;d\left(X,X_{i}\right)\neq 0\\ u_{i},\;\;\;\;\;\;\;\;d\left(X,X_{i}\right)=0 \end{array},$$
where $w_{i}\left(X\right)$ is the weight function, X is the interpolation point coordinate, Xi is the data point used for interpolation, and the Shepard method is used for the weighting calculation.

The purpose of the interpolation step is to calculate the distance from the unknown point to all points, to calculate the weighting of each point, and then to calculate the value at the interpolation point according to Equation (1).

The microbial determination results in the sample were calculated according to Equation (1), and the planar spatial distribution maps of the microorganisms in the sand layers at depths of 5, 10, 15, 20, 25, and 30 cm downward from the sand surface were drawn (for details, see Figure 9).

It can be seen that the distribution of microorganisms measured in each soil layer shows a high degree of spatial heterogeneity with a substantial edge effect. Based on the trend of the microbial biomass isoline and the corresponding microbial biomass value of each soil layer, it is obvious that the spatial heterogeneity at depths of 5–15 cm is largest, and that the distribution law is consistent with the profile of the average microbial value shown in Figure 10. It can also be seen in Figure 10 that, within the depth range of 5–15 cm in the sandy soil foundation, the square difference is much greater than that of other soil layers, indicating the greatest change of microbial activities.

Following the shaking table test, greater microbial-induced calcium carbonate mineralization was observed at the grouting hole in the bottom of the model box, and the grouting hole was largely blocked. This can also be visualized in the three-dimensional images of the microbial distribution (Figure 11). In Figure 11, the X and Y axes represent the length and width of the model box (cm), respectively, and the Z axis represents the number of microorganisms (×10^5^ cfu/mL). The amplitude of the peaks and valleys shows the heterogenous distribution of the microorganisms in the sandy soil foundation. It can also be seen that the numbers of microorganisms in the 5 and 10 cm sand layers are notably higher than in the other layers. Similar to the discussion on the spatial heterogeneity of the microbial distribution, maps of the distribution of calcium carbonate were drawn (Figure 12 and Figure 13), and the distribution profile of calcium carbonate is shown in Figure 10 (red line).

### 3.4. Analysis of the Causes of Heterogenous Distribution of Mineralization

It can be seen from the comparison of the test materials and grouting schemes that the density of both the bacterial solution and the nutrient salts is much lower than that of the saturated sand (the relative density is 40%, and the saturated solution is hydroxymethyl propyl cellulose). The pressure difference between the two also increased with the increasing depth during the grouting process. Moreover, the boreholes on the pipe used for distributing the grouting fluid were all buried 3 cm below the surface of the sand layer. Owing to the pressure of the peristaltic pump on the flow of grouting fluid in the pipe, the closer the pipe is to the surface of the sand foundation, the easier it is for the grouting fluid to flow out. Accordingly, the pressure immediately decreases after the fluid flows out, resulting in a greater concentration of the microbial grouting fluid in the upper surface layer in comparison with the middle and lower layers.

A comparison of Figure 9 and Figure 12 reveals that mineralization is strongest near sites with more microorganisms in the sand layer and weakest in the lower layer of the box. One reason for this distribution is that the formation of calcium carbonate is mainly realized by extracellular enzymes secreted by microorganisms [34]. In terms of body size, extracellular enzymes are nanoscale and microorganisms are micrometer scale. This substantial difference between the two makes diffusion easier for extracellular enzymes, and therefore the diffusion speed of extracellular enzymes is faster than that of microorganisms. Thus, extracellular enzyme mineralization will occur in places that are unreachable to microorganisms, thus directly leading to weak mineralization in sand samples with large depth.

In terms of the biological reinforcement test, it is important to resolve the problem of consistency in the relative compactness of the falling sand layer in the model. It is also important to ensure uniformity of the methylcellulose saturated solution in the porous medium (i.e., the sand is sufficiently saturated by the reverse pressure) because these affect the success or failure of the test.

## 4. Conclusions

The medium-strengthening microbial reinforcement scheme substantially improves performance in the prevention of liquefaction of sand (for a foundation of approximately 9.6 m) and improves the shear stiffness of sand to a certain extent.The type of calcium salt has an important influence on the microbial treatment of liquefied sand. Under the same conditions, Ca(CH_3_COO)_2_ is considered most suitable for the treatment of liquefied sand foundations.The process and effect of microbial grouting used to reinforce a liquefied sand foundation are complex and uncertain. The spatial heterogeneity of the microbial density is particularly important at depths of 5–10 cm. The mineralization effect of the induced calcium carbonate is optimal at depths of 15–20 cm in sandy soil and shows obvious spatial heterogeneity with a substantial edge effect.Although microbial mineralization can effectively improve the liquefaction resistance of saturated sand, a sudden change in the process of calcium carbonate deposition will alter the cementation of sand with depth, resulting in obvious complexity and uncertainty in the curing strength, which directly affects the shear stiffness of the soil under dynamic loads and constitutes one of the reasons for the degradation of shear stiffness of sand during liquefaction.For the biological-reinforcement test, it is important to overcome the problem of consistency in the relative compactness of the falling sand layer in the model and to ensure uniformity of the methylcellulose saturated solution in the porous medium (i.e., the sand is sufficiently saturated by reverse pressure).

## 5. Discussion

Zeghal and Elgamal [35] proposed a method for analyzing liquefaction by field measured seismic waves and developed a formula for calculating the shear strain and shear stress of soil based on the measured acceleration-time history, which has been popularized and applied to dynamic centrifuge tests. The typical liquefaction curves of sandy soil presented in this paper show that shear stiffness will decrease and that degradation will be severe.

Figure 10 shows a sudden change in calcium carbonate deposition near the depth of 20 cm in the microbial grouting model, and that the calcium carbonate increases slightly from this point to the bottom of the model box. Although the difference is not significant, there is a notable difference in the solidification and cementation of loose sand via biological mineralization on both sides of this abrupt change point, and there is a difference in the shear stiffness of the soil under seismic load. This is consistent with Zeghal and Elgamal [35], who proposed that the shear stiffness of soil in the typical liquefaction process of sand soil degenerates violently.

2.Microbial mineralization of saturated sand has obvious uneven distribution and a significant edge effect. The treatment effect would be further improved by reasonably adjusting the batch of microbial grouting, optimizing the length of the grouting pipe, and optimizing the distribution of the injection port on the basis of the actual status of the liquefiable sand in the engineering application.

## Figures and Tables

**Figure 1 materials-15-06102-f001:**
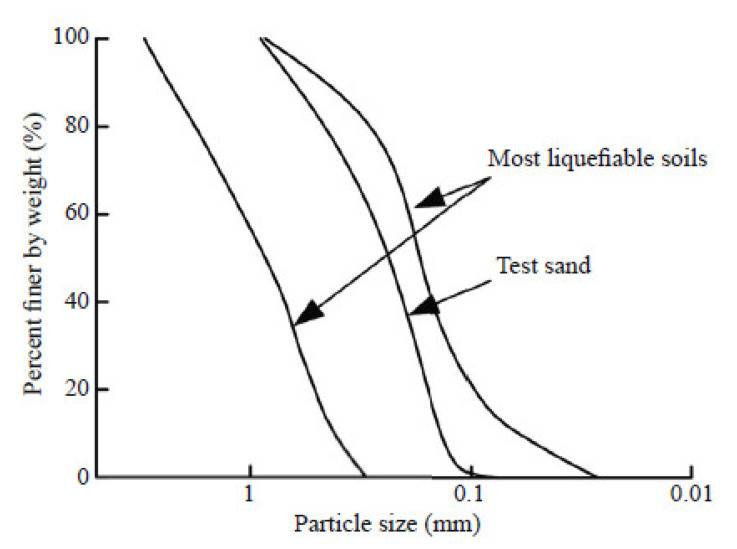
Particle size distributions of the XiFeng standard sand.

**Figure 2 materials-15-06102-f002:**
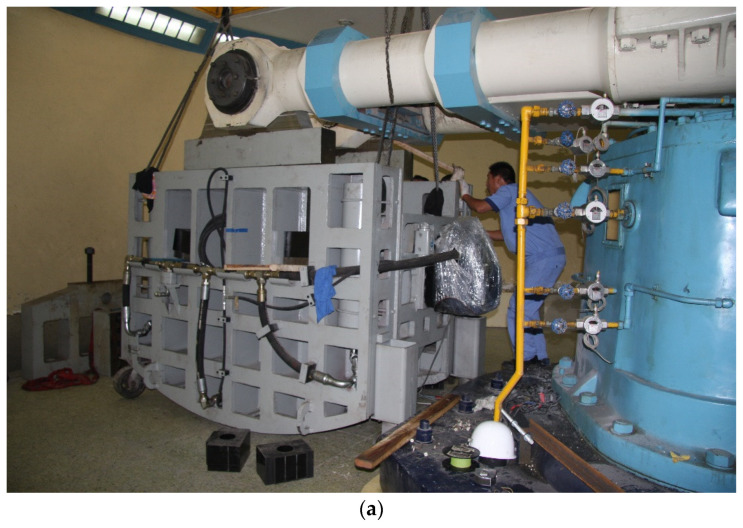
Geotechnical centrifuge and model box. (**a**) Lxj-4-450 geotechnical centrifuge with vibrating table. (**b**) ESB equivalent shear model box.

**Figure 3 materials-15-06102-f003:**
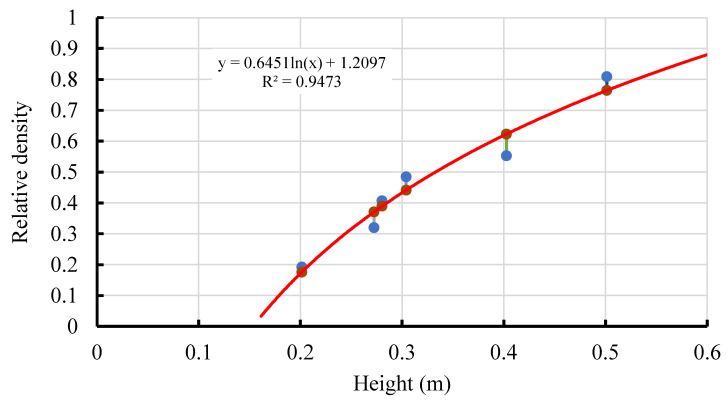
Relationship between the sand’s falling distance and the dry sand’s relative density.

**Figure 4 materials-15-06102-f004:**
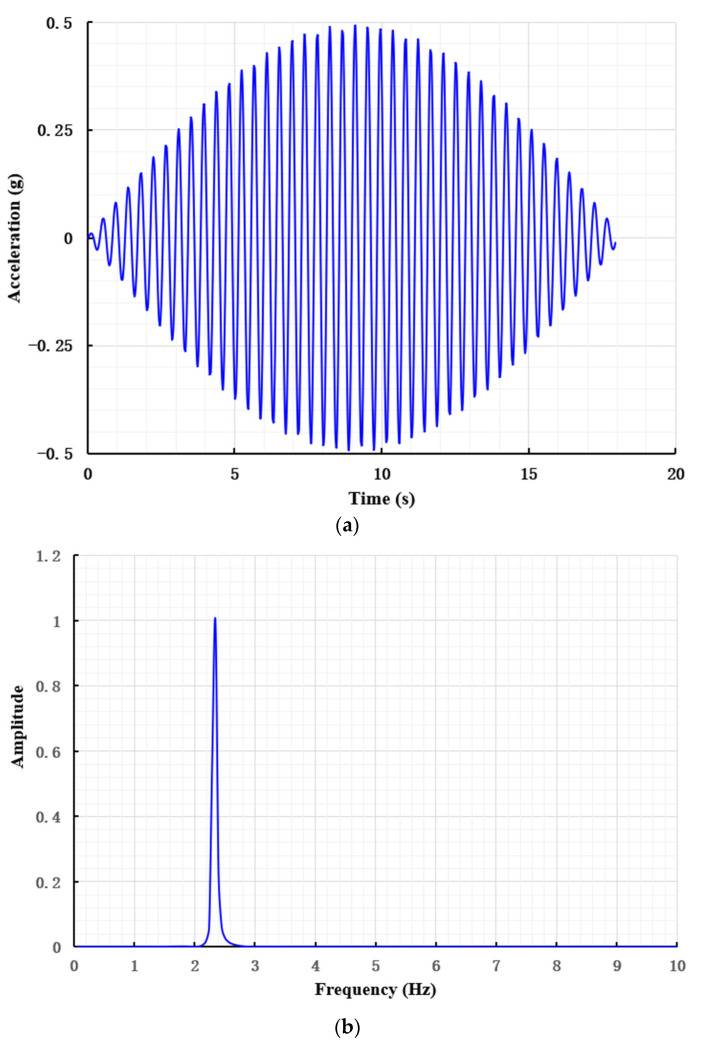
Frequency domain response of sine wave acceleration amplitude and loading signal. (**a**) Acceleration amplitude of loading signal (prototype). (**b**) Frequency domain response of normalized loading signal (prototype).

**Figure 5 materials-15-06102-f005:**
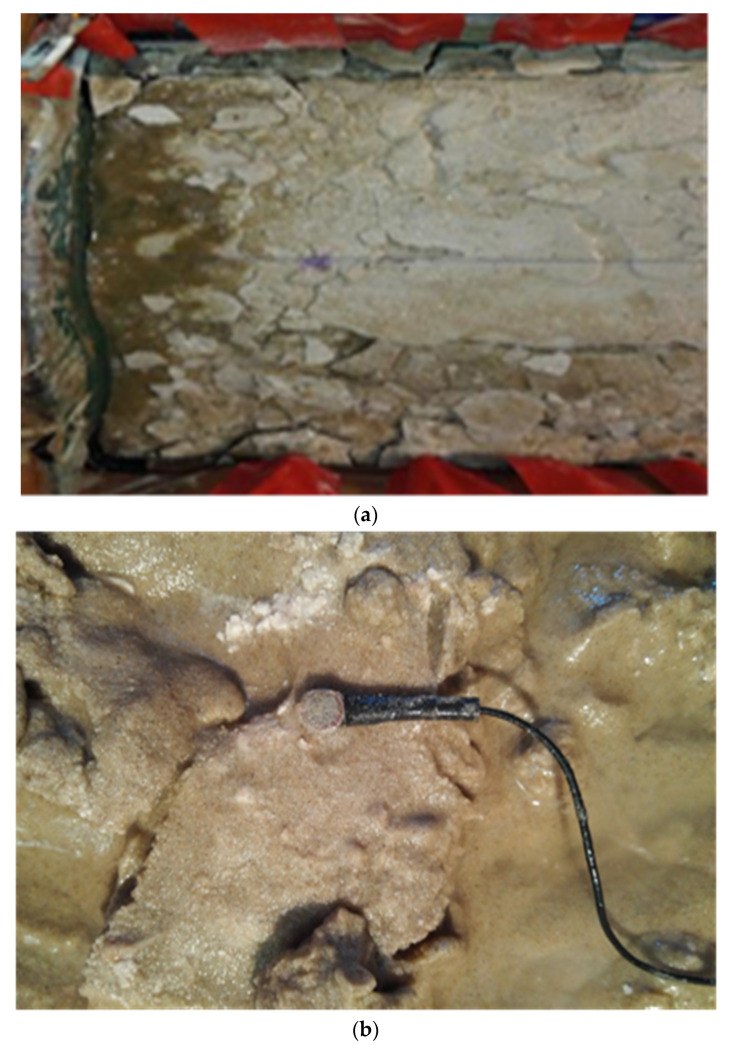
Metallogenic surface of the sandy soil. (**a**) Fragmentation of the surface calcium carbonate layer. (**b**) Sensor installation.

**Figure 6 materials-15-06102-f006:**
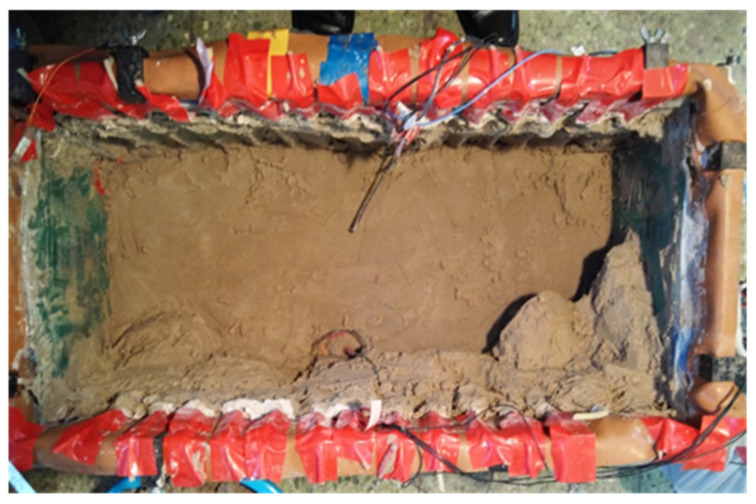
Spatial heterogeneity of sandy soil mineralization.

**Figure 7 materials-15-06102-f007:**
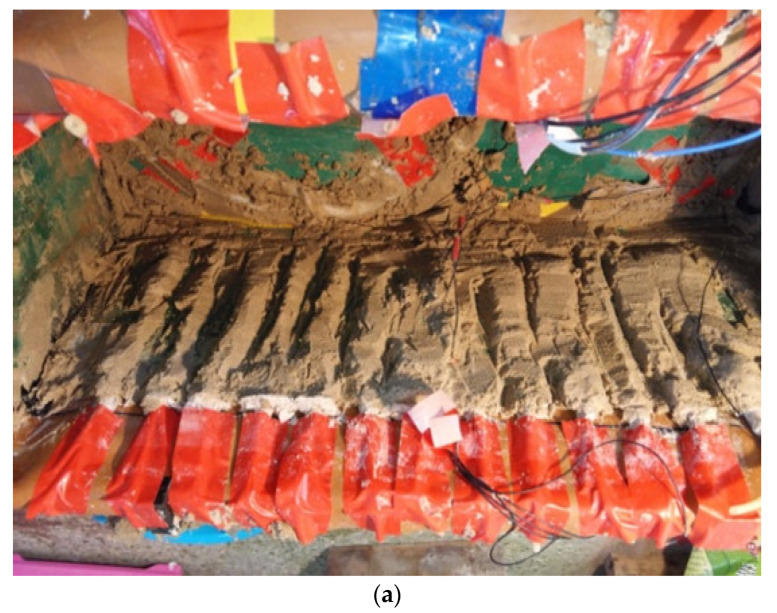
Heterogeneity of calcium carbonate deposition in the sand in the horizontal plane. (**a**) Obvious calcium carbonate deposition near the bored pipe. (**b**) Calcium carbonate blockage at the injection port of the bored pipe.

**Figure 8 materials-15-06102-f008:**
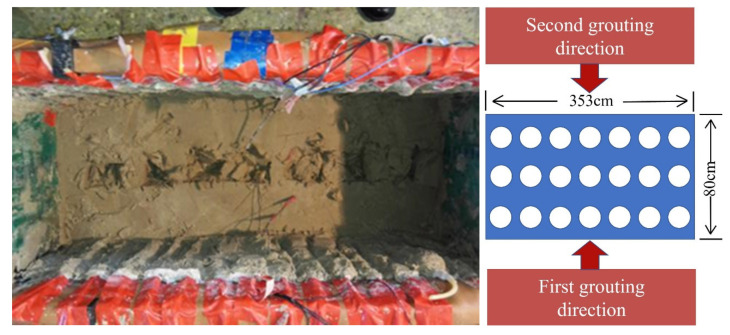
Schematic of the grid sampling method.

**Figure 9 materials-15-06102-f009:**
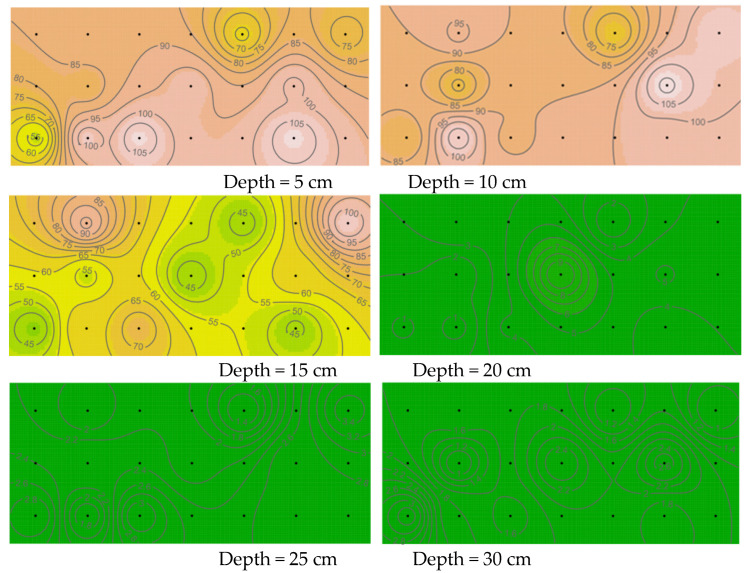
Spatial distribution of microorganisms in the different layers of the sandy soil (unit: 10^5^ cfu/mL).

**Figure 10 materials-15-06102-f010:**
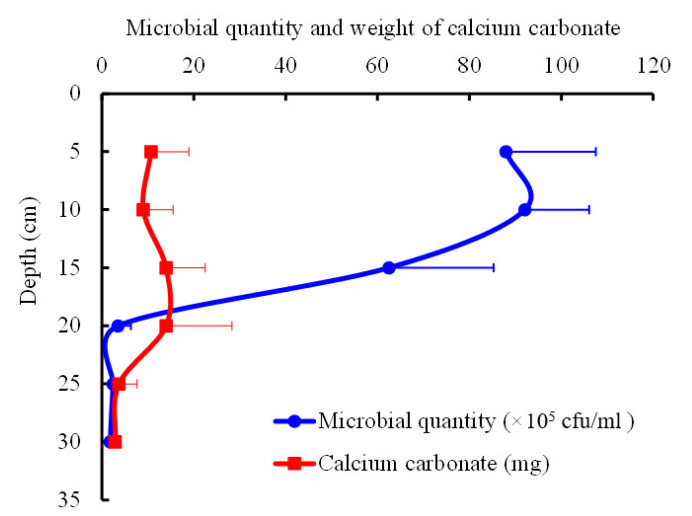
Distribution profiles of microorganisms and calcium carbonate in the sandy soil.

**Figure 11 materials-15-06102-f011:**
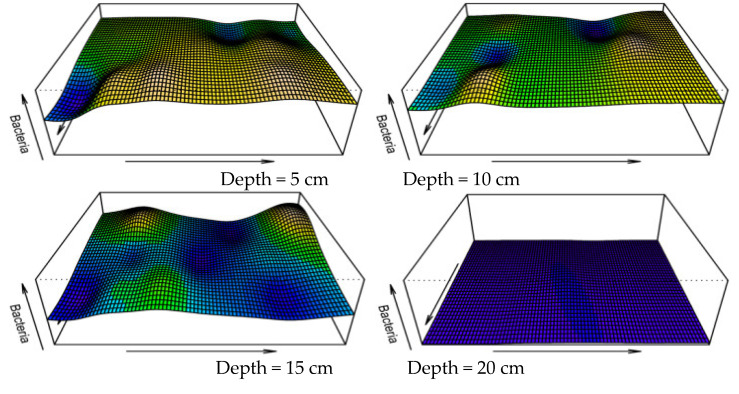
Three-dimensional distribution of microorganisms in the sandy soil.

**Figure 12 materials-15-06102-f012:**
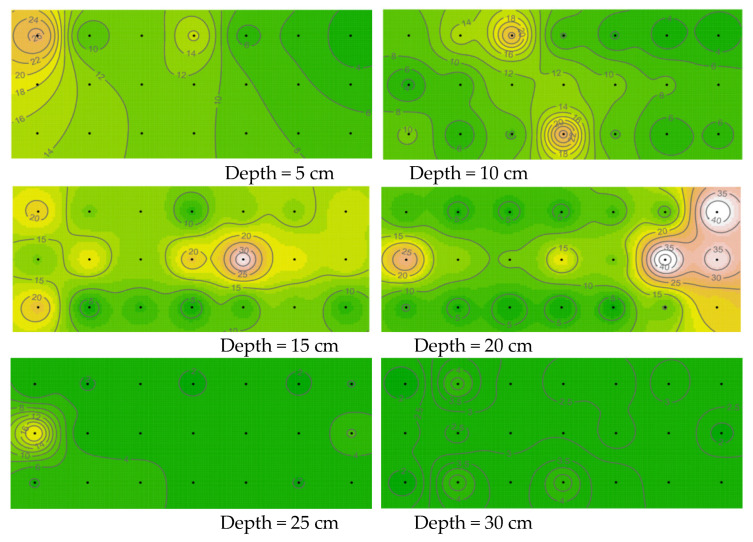
Spatial distribution of calcium carbonate in the different sand layers of the sandy soil.

**Figure 13 materials-15-06102-f013:**
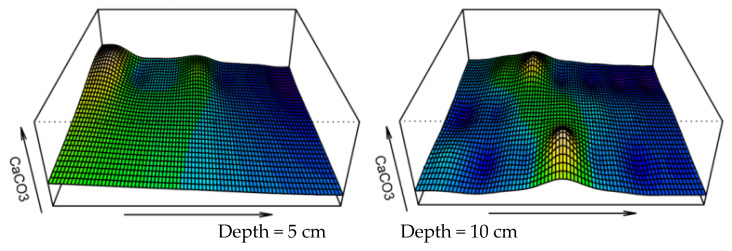
Three-dimensional distribution of calcium carbonate in the sandy soil foundation.

**Table 1 materials-15-06102-t001:** Physical properties of the XiFeng standard sand.

Material	*ρ*_min_ (g/cm^3^)	*ρ*_max_ (g/cm^3^)	*e* _min_	*e* _max_	*e*	*D*_r_ (%)	*d*_50_ (mm)
Standard sand	1.343	1.621	0.624	0.961	0.776	40	0.158

**Table 2 materials-15-06102-t002:** Modeled error of relationship curve of falling distance and relative compaction.

Height (m)	0.201	0.272	0.280	0.304	0.403	0.501
Relative density measured value (%)	19.188	31.980	40.660	48.426	55.279	80.863
Relative density fitted value (%)	17.584	37.081	38.924	44.157	62.292	76.432
Modeled error	1.604	−5.102	1.736	4.270	−7.013	4.431

## Data Availability

The data sets supporting the results of this article are included within the article.

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
