# Peer review of "Spatial Distribution Characteristics of Microbial Mineralization in Saturated Sand Centrifuge Shaking Table Test"

_materials, 2022, doi:10.3390/ma15176102_

Round 1
Reviewer 1 Report
The document “Spatial distribution characteristics of microbial mineralization in saturated sand centrifuge shaking table test” is interesting and provides relevant information. Small details should be attended to before publication. Consider these comments:
Relevant and appropriate abstract and introduction.
Materials and methods are well described.
I liked the description of the inverse distance weighted spatial interpolation method. Very educational and complete.
Add the modeled error of the parameter x to the inset equation of Figure 3.
Improve the resolution of Figure 4.
What is the title of the abscissa in Figure 10?
Reviewer 2 Report
The manuscript “Spatial distribution characteristics of microbial mineralization in saturated sand centrifuge shaking table test” " is a good contribution
However, the following comments are made so that the manuscript can be considered for publication:
· Add a paragraph where the type of microorganisms that induce the formation of calcium carbonate crystals. Also add, which are the most suitable conditions so that it can be generated. · Improve the quality of figures. · It is recommended to evaluate different concentrations of microorganisms and how it affects the time since they were added.
· Why is a higher concentration of calcium carbonate generated at depths of 15 and 20 cm? (figure 10).
Reviewer 3 Report
The submitted manuscript entitled as "Spatial distribution characteristics of microbial mineralization in saturated sand centrifuge shaking table test" presents a rather comprehensive and systematic study . However, there are some issues that need to be addressed prior to publishing in the journal as follows:
The introduction part is too short. More information is needed in the Introduction part.
The introduction part looks like thesis data.
Authors used basic (basal) medium for their sporulation experiments, kindly cite the reference for basic medium
A lack of discussion about the results. Authors have not justified their results with previous research.
Improve the quality of figures.
The conclusion does not provide much details about future recommendations.
The paper is not written in a scientific way.
Lots of grammatical errors.
The figures are too small and not clear.
References are too old.
Round 2
Reviewer 3 Report
The said changes have been incorporated.